# Plasmon Effect of Ag Nanoparticles on TiO_2_/rGO Nanostructures for Enhanced Energy Harvesting and Environmental Remediation

**DOI:** 10.3390/nano13010065

**Published:** 2022-12-23

**Authors:** Seenidurai Athithya, Valparai Surangani Manikandan, Santhana Krishnan Harish, Kuppusamy Silambarasan, Shanmugam Gopalakrishnan, Hiroya Ikeda, Mani Navaneethan, Jayaram Archana

**Affiliations:** 1Functional Materials and Energy Devices Laboratory, Department of Physics and Nanotechnology, SRM Institute of Science and Technology, Kattankulathur, Chennai 603 203, India; 2Research Institute of Electronics, Shizuoka University, 3-5-1 Johoku, Naka-Ku, Hamamatsu 432-8011, Japan; 3Nanotechnology Research Center (NRC), SRM Institute of Science and Technology, Kattankulathur, Chennai 603 203, India

**Keywords:** solar energy, dye degradation, surface plasmon resonance effect, TiO_2_/rGO/Ag, hybrid nanostructures

## Abstract

We report Ag nanoparticles infused with mesosphere TiO_2_/reduced graphene oxide (rGO) nanosheet (TiO_2_/rGO/Ag) hybrid nanostructures have been successfully fabricated using a series of solution process synthesis routes and an in-situ growth method. The prepared hybrid nanostructure is utilized for the fabrication of photovoltaic cells and the photocatalytic degradation of pollutants. The photovoltaic characteristics of a dye-sensitized solar cell (DSSC) device with plasmonic hybrid nanostructure (TiO_2_/rGO/Ag) photoanode achieved a highest short-circuit current density (*J_SC_*) of 16.05 mA/cm^2^, an open circuit voltage (*V_OC_*) of 0.74 V and a fill factor (*FF*) of 62.5%. The fabricated plasmonic DSSC device exhibited a maximum power conversion efficiency (PCE) of 7.27%, which is almost 1.7 times higher than the TiO_2_-based DSSC (4.10%). For the photocatalytic degradation of pollutants, the prepared TiO_2_/rGO/Ag photocatalyst exhibited superior photodegradation of methylene blue (MB) dye molecules at around 93% and the mineralization of total organic compounds (TOC) by 80% in aqueous solution after 160 min under continuous irradiation with natural sunlight. Moreover, the enhanced performance of the DSSC device and the MB dye degradation exhibited by the hybrid nanostructures are more associated with their high surface area. Therefore, the proposed plasmonic hybrid nanostructure system is a further development for photovoltaics and environmental remediation applications.

## 1. Introduction

The significant increase in energy requirements and the depletion of fossil fuels have caused researchers to develop energy harvesting from renewable energy resources [1,2]. The use of solar-driven photovoltaic technologies and heterogeneous photocatalysis offers an appealing solution to the current global energy crisis and environmental remediation [3,4,5]. In terms of solar energy conversion technologies, dye-sensitized solar cells (DSSC) and heterogeneous photocatalytic dye degradation are the most attractive and promising areas of research to address energy and environmental concerns [6,7]. Semiconducting titanium dioxide (TiO_2_) nanostructures are typically potential candidates as photoanode materials in DSSC devices and photocatalysts for the heterogenous photodegradation of toxic dyes due to their unique physicochemical properties [8,9,10]. However, the wide bandgap of the TiO_2_ nanostructures (3.2–3.3 eV) has the absorption region below 4% in the entire solar spectrum compared to that of the visible region (43% solar energy) [11]. As a result, inefficient use of the visible light spectral portion and the fast recombination of electron-hole pairs (TiO_2_) are significant constraints in the large-scale development of efficient photoanodes for DSSC devices and photocatalysts [12,13,14]. To overcome these constraints, different TiO_2_ nanostructure composites with two-dimensional (2D) carbon materials have been employed as an effective strategy in recent decades. In nature, 2D-single molecular layered structures with sp^2^ hybrid carbon atoms, i.e., graphene, possesses high surface area (ca. 2600 m^2^/g), electron mobility (ca. 15,000 m^2^/V.s at room temperature) and unrestricted movement of electrons in the crystal lattice [2,15,16]. During the formation of the nanocomposite, TiO_2_ nanostructures are bonded with the graphene surface due to the presence of intermolecular forces. This increases the number of electron spots and electron bridges, which promotes electron transport with a suppressed recombination rate at the interface [17]. Moreover, reduced graphene oxide (rGO) nanosheet composites with TiO_2_ nanostructures offer enhanced active surface area, good electrical conductivity and a lower recombination rate of photon-induced-charge carriers for photovoltaic and photocatalytic performance [18]. Manikandan et al. reported TiO_2_ along with rGO not only enhanced the surface area but also influenced the short-circuit current (*J_SC_*) in a device due to the high carrier mobility behavior of the rGO [19].

In recent decades, the localized surface plasmon resonance (LSPR) phenomenon of noble metal nanoparticles has played a dual role as potential visible sensitizers and electron sinks in the degradation of pollutant dyes and in DSSC device performance. It is a well-known strategy to trap electrons and harvest maximum light for the development of high-efficiency photovoltaic devices. Amine-functionalized TiO_2_ composites with GO and Ag nanoparticles exhibited high current density due to the improved electron transfer at the photoanode/electrolyte interface, as reported by Kandasamy et al. [20]. The incorporation of Ag nanoparticles (NPs) into TiO_2_-carbon nanotube (CNT) nanocomposites exhibited a photocatalytic activity of 66% for methylene blue (MB) degradation under visible light due to the presence of the CNTs and Ag NPs, as reported by Zhao et al. [21]. MB is a well-known, highly carcinogenic thiazine pollutant that has been manufactured and used in a variety of industries for various purposes. Therefore, it is strongly recommended to remove such a persistent contaminant from any given aqueous solution [22].

Moreover, the introduction of Ag NPs onto a TiO_2_ nanocomposite improved the absorption-coefficient of the organic dye and eventually enhanced the optical absorption in the visible-light region [23,24]. It is similarly promising to achieve superior electrical conductivity with a prolonged lifetime of the photogenerated charge carriers for rGO/TiO_2_ nanocomposites [25]. For example, Huan et al. prepared a flower-shaped nanosheet rGO/TiO_2_ composite material that exhibited a photocatalytic efficiency of 92.3% under UV–visible light irradiation for the degradation of a rhodamine B (RhB) solution [26]. Duygu et al. reported the development of rGO-TiO_2_-CdO-ZnO-Ag based composites that exhibited an excellent degradation rate of methylene blue (MB) dye (15 min) with 91% photocatalytic efficiency under UV light irradiation [27]. Similarly, Zohreh et al. reported the influence of surface plasmon resonance on the photovoltaic characteristics of Ag/TiO_2_ in a photoanode-based DSSC device with a power conversion efficiency (PCE) of 6.5% under 1 sun simulated solar irradiation [28]. As a result, the combination of the LSPR influence of Ag NPs and the conductivity of rGO nanosheets with the reduced recombination rate of the TiO_2_/rGO/Ag hybrid nanostructures is advantageous for long-term dye degradation photocatalysts and photoanodes for DSSC devices.

In the present work, we report the in situ growth synthesis of mesosphere TiO_2_/rGO nanosheets/Ag NPs as a plasmonic hybrid nanostructures for visible-light-responsive DSSCs and photocatalytic applications. The prepared hybrid nanostructures are further examined with respect to their structural, morphological and optical properties using various advanced characterization techniques. The influence of the LSPR properties on a constructed DSSC device with the visible-light-driven photoanode and photocatalyst dye (MB) degradation is systematically investigated with an appropriate mechanism.

## 2. Experimental Section

### 2.1. Materials and Reagents

Titanium tetra-isopropoxide (TTIP), ethylene glycol (C_3_H_6_O_2_), acetone (C_3_H_6_O), ethanol (C_2_H_6_O), graphite powder, sodium nitrate (NaNO_3_), sulfuric acid (H_2_SO_4_), potassium permanganate (KMnO_4_), hydrochloric acid (HCI), hydrogen peroxide (H_2_O_2_), isopropyl alcohol (IPA), 1,4-benzoquinone (BQ), ethylenediaminetetraacetic acid (EDTA) and silver nitrate (AgNO_3_) were purchased from SRL Co., Mumbai, India. All the purchased chemicals were analytical grade and used in the synthesis without further purification.

### 2.2. Preparation of TiO_2_ Mesospheres

TiO_2_ mesospheres were prepared through a combined route of sol-gel and solvothermal synthesis. In the first step (sol-gel), 3 mL of TTIP was slowly added into 150 mL of ethylene glycol, and the solution was continuously stirred for 12 h at room temperature. After a 12 h stirring process, 300 mL of acetone with 2 mL of de-ionized water (DI) were added into the solution and stirring was continued for 2 h to obtain a white suspension. Subsequently, the white suspension was collected and subjected to several centrifugations with ethanol and DI water, respectively. The obtained product was dried at 60 °C for 10 h to remove the impurities and form titanium glycolate. In the second step of the procedure (solvothermal route), 1 g of titanium glycolate was dispersed into 60 mL of a mixed solvent of ethanol and DI water under stirring. The prepared white solution was then transferred into a 100 mL autoclave and kept at 180 °C for 12 h. Finally, the TiO_2_ mesospheres were obtained after annealing at 350 °C for 1 h.

### 2.3. Preparation of Graphene Oxide (GO)

Graphene oxide (GO) was prepared under room-temperature conditions by the oxidation of natural graphite powder using a modified Hummer’s method [29]. Briefly, 1 g of natural graphite powder and 0.5 g of sodium nitrate were blended in 23 mL of concentrated sulfuric acid under vigorous stirring for 30 min. A total of 3 g of well-ground potassium permanganate was then slowly added into the mixed solution and stirred for 30 min under ice-bath conditions at 7 °C. The mixed solution was then stirred at 35 °C for 30 min. To quench the vigorous oxidation process in the solution, 3 mL of hydrogen peroxide was added to 60 mL of DI water, and the solid GO powder was obtained after washing several times with 5% of HCL and DI water, then dried overnight at 60 °C.

### 2.4. Preparation of Mesosphere TiO_2_/rGO Sample

Prepared TiO_2_ mesospheres (0.2 g) were added to a mixed solution of an equal portion of DI water (15 mL) and ethanol (15 mL) (as solution A). Then, 30 mg of GO powder was dispersed in the same ratio of solvent (DI and ethanol) under stirring for 1 h (as Solution B). Solutions A and B were then mixed with continuous stirring for 1 h. The resultant mixed solution was transferred into a 100 mL autoclave and kept in hot air at 180 °C for 12 h. The final product of mesosphere TiO_2_/rGO was obtained by washing the product with DI water and drying overnight at 60 °C.

### 2.5. Preparation of TiO_2_/rGO/Ag Hybrid Nanostructure by In Situ Growth

The TiO_2_/rGO/Ag hybrid nanostructures were synthesized by an in situ hydrothermal process. In this procedure, 10 mg of silver nitrate was dissolved in 60 mL of DI water (30 mL) and ethanol (30 mL), and then 0.2 g of the prepared TiO_2_/rGO was added to the above solution under continuous stirring for 1 h. The net solution was transferred into an autoclave and kept at 180 °C for 4 h. The final product was washed with DI water and dried at 60 °C.

### 2.6. Characterization

Structural analysis was conducted using X-ray diffraction (XRD; PANalytical, Malvern, UK) with Cu Kα radiation (λ = 1.5406 Å) in the 2-theta range between 10° and 80° with a scanning rate of 0.02°/min. Raman spectra of the prepared sample were obtained using a micro-Raman spectrometer (LABRAM HR Evolution, Horiba, Longjumeau, France) with an excitation wavelength of 532 nm. The surface morphology of the prepared samples was analyzed using high-resolution scanning electron microscopy (HR-SEM; Apreo S, Thermo Fisher Scientific, Hillsboro, OR, USA) with an acceleration voltage of 15 kV. Further analysis of the surface morphology was conducted using high-resolution transmission electron microscopy (HR-TEM; JEM-2100, JEOL, Tokyo, Japan) with an acceleration voltage of 200 kV to reveal the atomic interplanar morphology and the elemental composition of the prepared samples. Ultraviolet-visible diffuse reflectance spectroscopy (UV-DRS, V-750, JASCO, Tokyo, Japan) measurements were conducted in the range of 200 nm to 800 nm. To analyze the emission properties of the prepared sample, photoluminescence (PL; FP8600, JASCO, Tokyo, Japan) spectra were measured at room temperature. The surface area and pore size distribution of the samples were characterized by the BET (Brunauer–Emmett–Teller) and BJH (Barrett–Joner–Halenda) methods (Autosorb IQ series, Quantachrome Instruments, Boynton Beach, FL, USA). X-ray photoelectron spectroscopy (XPS) was performed via a Kratos analytical instrument (ESCA 3400, Shimadzu Corporation, Kyoto, Japan). The percentage of mineralization efficiency was determined from total organic carbon (TOC) measurements (TOC-L, Shimadzu, Kyoto, Japan). Photovoltaic characterization of the fabricated devices was performed using a solar simulator (Sciencetech, Class A, Lamp: 300 W, London, ON, Canada). The I-V measurement and incident photocurrent efficiency (IPCE) of fabricated devices was measured using the same solar simulator over the wavelength range of 200 nm to 800 nm.

### 2.7. Photocatalytic Experiments

The photocatalysts measurement were carried out in our laboratory, SRM Institute of Science and Technology, Chennai (28°4′ N; 82°25′ E), in April 2020 (8 April to 10 May). Daylight from 9 am to 12 pm was utilized to perform the photocatalytic experiment with an average light intensity of 68.2~89.4 mW/cm^2^. The photocatalytic properties of the as-synthesized samples were performed using MB dye as a model pollutant. In a typical photocatalytic reaction, 10 ppm of MB dye was added to 50 mL of DI water and stirred for 5 min. The solution was maintained in the dark for 20 min under stirring to achieve an adsorption–desorption equilibrium. At regular time intervals of the photocatalytic dye degradation reaction solution (20 min), 3 mL aliquots of the solution were sampled and UV–Vis spectra were measured.

### 2.8. DSSC Device Fabrication

Prepared samples, such as TiO_2_, TiO_2_/rGO and TiO_2_/rGO/Ag (0.25 g each), were dispersed in 2 mL of stock solution (prepared by mixing an equal amount of DI water and ethanol) and ground for 15 min. To obtain a paste-like formation, 2 mL of the stock solution was mixed with 200 µL of acetic acid and 100 µL of surfactant (Triton 100-X) during the grinding process. The resultant colloidal solution was then uniformly coated on a fluorine-doped tin oxide (FTO) substrate with optimized conditions using a doctor blade technique. After drying at 100 °C for 10 min, the coated substrate was annealed at 450 °C for 30 min. The coated FTO was then immersed in dye solution for 12 h, which consisted of 0.03 M dis-tetrabutylammonium cis-bis(isothiocyanato)bis(2,2″-bipyridyl-4,4′-dicarboxylato) ruthenium (II) (N719). The prepared photoanode was subsequently clamped with a Pt-coated counter electrode to form a sandwich type of device. An electrolyte solution consisting of 0.6 M dimethylpropylimidazolium iodide, 0.1 M lithium iodide, 0.01 M iodine and 0.5 M 4-tert-butylpyridine in acetonitrile was then filled in between the layers of the sandwich-type device.

## 3. Results and Discussion

### 3.1. Structural and Compositional Analysis

Figure 1a shows the XRD patterns of prepared samples of pristine GO, mesosphere TiO_2_, TiO_2_/rGO and TiO_2_/rGO/Ag. A strong and sharp diffraction peak is observed at 9.8°, which corresponds to the (0 0 1) crystal plane of GO. Using Bragg’s equation, the interlayer distance of the as-prepared GO sheet was estimated to be 0.9 nm, whereas graphite powder shows an interlayer distance around 0.33 nm [30]. The XRD pattern obtained for GO indicates that the bulk graphite is successfully reduced as rGO nanosheets [31]. Furthermore, the characteristic peaks of TiO_2_ are observed at 25.28°, 37.97°, 47.95°, 53.84°, 55.02°, 62.40°, 68.70° and 75.20°, which are assigned to the (1 0 1), (0 0 4), (2 0 0), (1 0 5), (2 1 1), (2 0 4), (1 1 6) and (2 1 5) planes, respectively. In the case of TiO_2_/rGO/Ag, the peaks observed at 38.12°, 44.26°, 64.33° and 77.35° are associated with the cubic phase of the Ag (1 1 1), (2 0 0), (2 2 0) and (3 1 1) planes (JCPDS: 04-0783). Therefore, the obtained XRD pattern of the TiO_2_/rGO/Ag composite reveals the coexistence of rGO, TiO_2_ and Ag materials and confirms the effective formation of a hybrid nanostructure.

To explore the structural characteristics, Raman spectroscopy was performed for GO, TiO_2_, TiO_2_/rGO and TiO_2_/rGO/Ag samples, as shown in Figure 1b. The obtained Raman spectrum of pristine GO revealed the D band at 1350 cm^−1^ due to the disordered nature of the graphene structure with sp^3^ defects (as given in Appendix A). In addition, the G band (at 1587 cm^−1^) was observed in the Raman spectrum of pristine GO, which was attributed to the in-plane vibrations of C–C stretching in graphitic materials, as well as the doubly degenerate phonon mode in the Brillouin zone [32]. As shown in the inset image of Figure 1b, the G band shifted from 1587 to 1599 cm^−1^, which is direct evidence of the effective reduction of GO to rGO with a constant D band for TiO_2_/rGO [33]. In comparison with pristine GO, the I_D_/I_G_ intensity ratio of the TiO_2_/rGO and TiO_2_/rGO/Ag samples was slightly reduced to 0.83 from 0.87 (as shown in Appendix A). However, the reduction in the I_D_/I_G_ ratio indicates a considerable reduction in the sp^2^ domain size of carbon atoms, as well as the reduction of sp^3^ to sp^2^ [34]. In the case of TiO_2_/rGO/Ag, the Raman vibrational modes were observed at 147, 199, 396, 515 and 639 cm^−1^, which corresponds to the E_g_, E_g_, B_1g_, A_1g_ and E_g_ modes of anatase TiO_2_ [35]. On the other hand, the broadening of the E_g_ mode (147 cm^−1^) with a considerable peak shift revealed the presence of TiO_2_ mesospheres, as well as Ag NPs, on multilayer rGO. Furthermore, there were no significant changes in the observed Raman scattering modes of the plasmonic hybrid nanostructures compared with the hybrid sample (TiO_2_/rGO). Nevertheless, the single Raman characteristic peak of the anatase TiO_2_ phase was considerably enhanced due to the LSPR properties of Ag NPs on the TiO_2_/rGO/Ag hybrid surface [36]. The obtained Raman characteristic modes further confirm the successful formation of plasmonic hybrid nanostructures.

XPS measurements were performed to evaluate the chemical state and interaction of the prepared hybrid nanostructures and pristine GO samples. The surveyed spectra of TiO_2_, TiO_2_/rGO and TiO_2_/rGO/Ag indicate the presence of Ti, O, Ag and C elements without any impurities in the prepared hybrid nanostructures (as given in Appendix A). The deconvoluted C 1s spectrum of the as-prepared GO sample is shown in Figure 1c as follows: (i) At 282.55 eV, C bonds with sp^1^ carbon atoms [37], (ii) C=C bonds denote the aromatic sp^2^ structure groups present in the GO at 284.66 eV, (iii) the peak at 286.30 eV represents the carboxyl groups (C–O bonds) including epoxy and hydroxyl group [38]. In the case of the TiO_2_/rGO sample, the deconvoluted XPS peak of the C 1s spectrum shows the four diverse carbon bonds with various binding energies of 283.18 eV, 284.74 eV, 286.07 eV and 288.72 eV. Thus, the peak that appears at 283.18 eV indicates the chemical bonding between the C atom and the Ti atom (Ti–C) [34]. Furthermore, the peaks with reduced intensity emerges at 284.74 eV, 286.07 eV and 288.72 eV with extensively reduced intensity purely attributed to C=C, C–O, O=C–OH bonds as oxygenated functional groups, which highlights the formation of rGO from GO [39]. Appendix A shows the deconvoluted O 1s spectrum with two bands at 529 eV and 530 eV that correspond to the Ti–O–Ti and Ti–O–C groups. There is a slight peak shift in the TiO_2_/rGO sample from 530.95 eV to 531.12 eV, which indicates the strong binding of rGO with TiO_2_ mesosphere during the solvothermal process [40,41]_._ The chemical states of Ti species were thoroughly investigated using the XPS spectra, as shown in Figure 1e. The core level peak of Ti is split into a doublet peak observed at 458.40 eV and 464.10 eV, which corresponds to Ti 2p_3/2_ and Ti 2p_1/2_ with a splitting energy of 5.7 eV, mainly attributed to the Ti^4+^ state for the TiO_2_/rGO/Ag samples [42,43]. After infusing the Ag NPs into the hybrid nanostructure, the doublet peaks (Ti 2p_3/2_ and Ti 2p_1/2_) of the Ti 2p state have observed notable shifts to higher order, such as 458.40 eV to 458.51 eV (Ti 2p_3/2_) and 464.10 eV to 464.27 eV (Ti 2p_1/2_). As a result, a positive shift in the doublet peak of the Ti species directly reflects the alteration of the Fermi level, whereas the Ag species holds a lower Fermi level offset than that of TiO_2._ Therefore, the resultant Fermi level alignment in the plasmonic hybrid nanostructures are more favorable for fast electron transfer processes at the rGO/TiO_2_/Ag interface [44]. Figure 1f shows the doublet peaks of the Ag element at binding energies of 367.23 eV and 373.31 eV, which correspond to Ag 3d_5/2_ and Ag 3d_3/2_ with a splitting energy of 6.12 eV, which confirms the presence of the Ag NPs in the prepared TiO_2_/rGO/Ag hybrid nanostructures [45].

### 3.2. Morphological Analysis

The surface morphology features of GO nanosheets, TiO_2_ mesospheres, TiO_2_/rGO and TiO_2_/rGO/Ag hybrid nanostructures were investigated using HR-SEM, TEM and HR-TEM measurements, as shown in Figure 2. The transparent ultrathin GO nanosheets are visible in the HR-SEM and HR-TEM micrographs, as shown in Figure 2(a1–a3). As-prepared TiO_2_ mesosphere samples are agglomerate-free, uniformly distributed and spherical shaped, with an average size of ~570 nm (see Figure 2(b1)).

The surface morphology and atomic interplanar distance of prepared samples were investigated using TEM and HR-TEM, as shown in Figure 2(b2,b3). Figure 2(c1–c3) shows the uniform TiO_2_ mesospheres with an interplanar distance of 0.35 nm for the pure anatase TiO_2_ phase over the rGO nanosheet surfaces_._ The presence of Ag NPs embedded in the TiO_2_ mesosphere-coated rGO nanosheet surfaces is revealed by the TEM and HR-TEM images of the hybrid nanostructures. The interplanar distance of the prepared hybrid nanostructures was estimated from the HR-TEM image using a line profile analysis, as shown in Figure 2(d3). The calculated interplanar distance values were 0.35 nm and 0.23 nm for the anatase TiO_2_ crystal plane of (1 0 1) and the (1 1 1) crystal plane of the Ag NPs, respectively, which is well matched with the standard JCPDS cards (Nos. 21-1272 and 04-0783) [46]. Moreover, the elemental mapping and energy dispersive X-ray spectroscopy (EDX) spectrum of the prepared hybrid nanostructures indicate the coexistence of C, Ti, O and Ag elements without the presence of any impurities in the resultant hybrid sample (as shown in Appendix A). Therefore, the resultant surface morphology of the prepared hybrid ternary nanostructures indicates that the plasmonic Ag NPs are uniformly embedded into the TiO_2_ mesospheres anchored in the rGO nanosheet surfaces. Moreover, the thickness of the fabricated photoanode was estimated to be 14.2 µm (TiO_2_) and 13.8 µm (TiO_2_/rGO/Ag) and presented Appendix A.

### 3.3. Optical and Surface Area Analysis

UV-DRS was used to investigate the optical absorption of the prepared TiO_2_, TiO_2_/rGO and TiO_2_/rGO/Ag samples and is shown in Figure 3a. Due to the electronic transition of O 2p to Ti 3d, all of the prepared samples exhibit a typical optical absorption edge located at 392 nm of TiO_2_, which corresponds to a bandgap energy of 3.3 eV for TiO_2_ [47]. After the introduction of Ag NPs over the TiO_2_/rGO surface, the optical absorption of the hybrid nanostructure is significantly higher in the visible region compared to the other samples, due to the LSPR of Ag NPs on the TiO_2_/rGO surface [48]. Overall, the improved visible-light-region absorption observed for the prepared hybrid nanostructures indicates a greater number of photons harvested from the visible-light region, which is favorable for efficient photocatalysts and photoanodes for solar energy conversion devices.

PL emission spectroscopy measurements are widely used to understand the desired photon-induced charge separation properties of the prepared samples. PL spectra of the prepared samples with an excitation wavelength of 300 nm and in the range from 350 to 500 nm are shown in Figure 3b. A strong band edge emission peak is observed at around 389 nm for all the prepared nanostructure samples [49]. Another emission peak centered in the visible region of 480 nm indicates the presence of oxygen vacancies on the TiO_2_ surface. For the plasmonic hybrid nanostructures, the resultant emission peaks are significantly quenched, which is a direct indication of the reduction of the photon-induced electron-hole pair recombination rate that facilitates electron transfer at the TiO_2_/rGO/Ag interface [50,51]. The function group of GO and prepared materials were analyzed by FTIR spectrum [52,53] (Appendix A).

The surface area and pore size distribution of the prepared samples were evaluated using the BET and BJH methods shown in Figure 3c,d. In comparison, the specific surface area of Ag NP-based hybrid nanostructures have a surface area of 297.71 m^2^/g due to TiO_2_ mesosphere formation with the Ag NP-coated rGO nanosheets. The estimated surface areas of the TiO_2_ mesosphere and TiO_2_/rGO samples were 234.16 m^2^/g and 281.31 m^2^/g, respectively. As shown in Figure 3d, the estimated pore sizes were 2.15 nm, 1.52 nm and 1.91 nm for the TiO_2_, TiO_2_/rGO and TiO_2_/rGO/Ag hybrid nanostructures, respectively. The enhanced specific surface areas and pore size distributions of the hybrid nanostructures are more beneficial for the photoanodes in DSSC assemblies and photocatalysts for photocatalytic reactions.

### 3.4. Performance of DSSC Device

The influence of the LSPR properties of Ag NPs on the photovoltaic characteristics of a DSSC based on a TiO_2_/rGO/Ag photoanode under simulated sun irradiation with an AM 1.5 G filter was investigated. The current density vs. voltage characteristics of DSSC devices assembled with various photoanodes (TiO_2_, TiO_2_/rGO and TiO_2_/rGO/Ag) are given with error bars in Figure 4a. Compared with the TiO_2_ and TiO_2_/rGO photoanodes, the TiO_2_/rGO/Ag photoanode show superior photovoltaic performance with a maximal *J_SC_* of 16.05 mA/cm^2^ and an improved open circuit voltage (*Voc*) of 0.74 V, with a fill factor (*FF*) of around 62.50%. In addition, the hybrid-nanostructure-based plasmonic DSSC device achieved a higher power conversion efficiency of 7.27%, which is 1.7 times higher than that of a pristine TiO_2_ photoanode (4.01%). An overall comparison of a fabricated DSSC device’s photovoltaic characteristics are given in Table 1 and Appendix A. Figure 4b shows the IPCE of all of the fabricated DSSC devices in the range of 400 nm to 800 nm. The Ag-based DSSC device exhibited the maximum IPCE owing to the presence of N719 dye adsorption. The IPCE of the TiO_2_/rGO/Ag photoanode-based plasmonic DSSC device achieved a maximum value of ca. 77.82% at an incident wavelength of 550 nm due to improved photon harvesting efficiency and a higher number of electron extraction from the N719 dye molecule than the other devices [27,28,52]. The photovoltaic performance of the fabricated TiO_2_/rGO/Ag photoanode-based plasmonic DSSC cell has been compared with a recent report, as illustrated in Figure 4c [54,55,56,57,58,59]. A tentative operating mechanism of the proposed plasmonic DSSC device is illustrated to understand its enhanced photovoltaic performance. As shown in Figure 4d,e, upon 1 sun irradiation the electrons are excited from higher occupied molecular orbitals (HOMO) to lower occupied molecular orbitals (LUMO) levels of the dye molecule (N719). Under this irradiation, if the incident light coincides with the LSPR effect of Ag NPs (work function (WF) = −4.2 eV), then the electrons close to the Fermi level are excited into a higher energy state by receiving the energy from plasmon resonance, known as a hot electron, via non-radioactive process [60]. These electrons have sufficient energy to break the Schottky barrier formed between the TiO_2_ mesosphere (WF = −4.4 eV) and Ag NPs and can flow through the conduction band of the TiO_2_ mesosphere. Moreover, it is possible that this hot electron can be accepted and shuttled via rGO nanosheets (WF = −4.4 eV) [61,62,63]. The superior conductivity of rGO nanosheets could assist in accepting or transmitting electrons generated by TiO_2_ or Ag NPs to the appropriate photoanode. The hybrid nanostructure attained higher efficiency than the TiO_2_ mesospheres for the following reasons. (i) The large surface area provides high dye loading and efficiently scatters the incoming light within the device. (ii) The rGO nanosheets promote extraordinary charge transport where electrons come from TiO_2_ mesospheres or Ag NPs and limit the electron-hole recombination rate. (iii) The LSPR effect of the Ag NPs extends the light absorption range from the UV to the visible region.

Appendix A shows the transient photocurrent response of the TiO_2_/rGO/Ag composite under 1 sun light irradiation with the highest photocurrent density, which is approximately one-fold greater than that of TiO_2_/rGO. This confirms that the incorporation of Ag into the TiO_2_/rGO/Ag nanocomposite not only enhances the light harvesting but also significantly expedites the photon-induced charge carrier separation and transport properties of Ag NP-coupled hybrid composites. The transient photocurrent response of the TiO_2_/rGO/Ag composite was also performed under a UV filter, which confirmed the plasmonic Ag response photocurrent in the visible region. These results confirm that Ag could act as an acceptor of the photogenerated electrons by TiO_2_/rGO and encourage fast charge transportation due to the high metallic conductivity, which effectively suppresses the charge recombination in the composites.

### 3.5. Photocatalytic Performance

To demonstrate the photocatalytic MB dye degradation process, TiO_2_-, TiO_2_/rGO- and TiO_2_/rGO/Ag-based photocatalysts were employed under natural sunlight irradiation for three repetitions as shown in Appendix A. Figure 5a–c shows dye degradation profiles with MB as the pollutant with different irradiation times under natural sunlight for all of the prepared photocatalysts. The maximum absorption peak for the photocatalytic dye degradation process in the presence of a photocatalyst is displayed at around 664 nm due to the absorbance characteristics of MB dye. Under dark conditions, there were no significant changes in the UV–vis absorption spectra of all the prepared photocatalysts. The UV–vis absorption spectra of all the prepared photocatalysts were successfully recorded with a time interval of 20 min under continuous natural sun irradiation. The hybrid nanostructures of the TiO_2_/rGO/Ag photocatalyst exhibited a higher dye degradation efficiency of 93% under continuous natural sunlight irradiation for 160 min. Compared with pristine TiO_2_, the enhanced dye degradation efficiency of the plasmonic hybrid photocatalyst was almost 1.3 times higher due to the LSPR properties of the Ag NPs. On the other hand, the presence of rGO/Ag on the TiO_2_ surface creates a more favorable environment for fast photon-induced charge separation and transfer with an extended lifetime of charge carriers, whereas Ag NPs act as electron sinks for improved photocatalytic dye degradation [64].

To further understand the effect of dye degradation in the form of mineralization efficiency, TOC analysis of the TiO_2_/rGO/Ag photocatalyst was performed with respect to various time intervals, as shown in Figure 5e. After 160 min of natural solar irradiation, the mineralization efficiency of the TiO_2_/rGO/Ag photocatalyst in terms of carbon content removal in MB dye was approximately 80.35% (Figure 5e). The improved TOC of the plasmonic hybrid TiO_2_/rGO/Ag photocatalyst suggests that it efficiently mineralized MB dye into residual organic molecules, such as CO_2_ and H_2_O, in the MB dye solution. Furthermore, the photocatalytic MB dye degradation activity of TiO_2_/rGO/Ag was evaluated by the addition of various radical scavengers (Ag, EDTA, IPA and BQ) to determine the more active species e^−^, h^+^, OH^−^ and O^2−^ in the degradation system [65,66]. Figure 5f shows various scavenger photocatalytic MB dye degradation (MB) experiments with the TiO_2_/rGO/Ag system as a photocatalyst. For this study, 2 mg of various scavengers were mixed with MB dye solution in the presence of a prepared photocatalyst (15 mg) and exposed to natural sunlight for 160 min. The photocatalytic MB dye degradation efficiency of TiO_2_/rGO/Ag was significantly inhibited by 58.89% and 23.00% in the presence of IPA and BQ as scavengers, respectively. This reduced the photocatalytic dye degradation efficiency, and it reveals that OH^−^ and O^2−^ radicals are the main active species during the degradation of MB by TiO_2_/rGO/Ag under natural sunlight [67].

A possible mechanism for the photon-induced charge carrier separation and transfer in the prepared photocatalyst surface is depicted in Figure 6. The valence band (*E_VB_*) and conduction band (*E_CB_*) potentials of TiO_2_, TiO_2_/rGO and TiO_2_/rGO/Ag were estimated using the Mulliken electronegativity theory [53] and the following equations:(1)$$E_{VB}=x-E^{e}+0.5E_{g\;\;\;}$$
(2)$$E_{CB}=E_{VB}-E_{g\;}$$
where *x* is the electronegativity (5.81 eV for TiO_2_), *E_CB_* is the conduction edge potential, *E_VB_* is the valence band edge potential, *E^e^* is the free energy of the electrons in the reversible hydrogen scale (4.5 eV), and *E_g_* is the bandgap of the material. The *E_VB_* and *E_CB_* potentials of the TiO_2_/rGO/Ag system were estimated to be −0.34 eV and 2.96 eV, respectively, while *E_VB_* and *E_CB_* for pure TiO_2_ were estimated to be −0.32 and 2.94 eV, respectively.

Figure 6 shows the proposed photocatalytic mechanism for the natural-sunlight-responsive TiO_2_-RGO-Ag nanostructure. After natural sunlight irradiation, the Ag NPs’ dipolar characteristics of the SPR effect enable electrons generated from the Ag NPs to migrate to the CB of TiO_2_. In contrast, electrons in the CB of TiO_2_ could be transferred to the surface of the RGO nanosheets due to the favorable electron affinity of TiO_2_ and the lower WF of RGO. Further, it is postulated that the photogenerated electron from Ag particles can also be transferred into the RGO sheets. TiO_2_- and plasmon-excited Ag particles serve as electron transfer channels, which ensure charge separation efficiency. The oxidized Ag^+^ species accept electrons from water molecules (H_2_O) or OH^−^ adsorbed on the TiO_2_ surface or from the dye molecules present in the solution and are regenerated. The reaction with H_2_O or hydroxide ions produces hydroxyl radicals (OH). These radicals (O_2_ and OH) are influential oxidizing agents for the degradation of MB dye molecules. Therefore, the prepared ternary nanostructures offer superior photocatalytic dye degradation (MB) due to efficient photoinduced charge carrier separation [52,53].

## 4. Conclusions

In this study, TiO_2_/rGO/Ag hybrid nanostructures were successfully synthesized using a combination of solution processes and in situ growth and were then employed as photoanodes for DSSCs and as catalysts for photodegradation applications. The plasmon-enhanced DSSC devices demonstrate enhanced photovoltaic performance of 7.27% along with a higher short-circuit current of 16.05 mA/cm^2^ and an IPCE efficiency of 77.82% at 550 nm. The results suggest that the high photovoltaic performance of the plasmon-based TiO_2_/rGO/Ag device can be attributed to (i) the large specific area of TiO_2_/rGO/Ag, which leads to high dye loading; (ii) TiO_2_ mesospheres enhancing the light scattering effect of incoming light; and (iii) the incorporation of Ag NPs facilitating more induced photons and fast electron transport in the device. Upon natural sunlight irradiation, the prepared hybrid nanostructure shows an improved photocatalytic degradation of MB by 93% within 160 min, and the effects of different scavengers on the obtained photocatalytic activity were systematically investigated. The effects of optimum active surface area, the LSPR properties of Ag NPs and the enhanced electrical conductivity of the prepared ternary nanostructures combine to provide an enhanced visible-light-driven plasmonic DSSC device and photocatalyst for dye degradation (MB). The proposed plasmonic and hybrid-based nanostructures demonstrate an emerging strategy to establish large-scale applications of solar energy conversion technologies.

## Figures and Tables

**Figure 1 nanomaterials-13-00065-f001:**
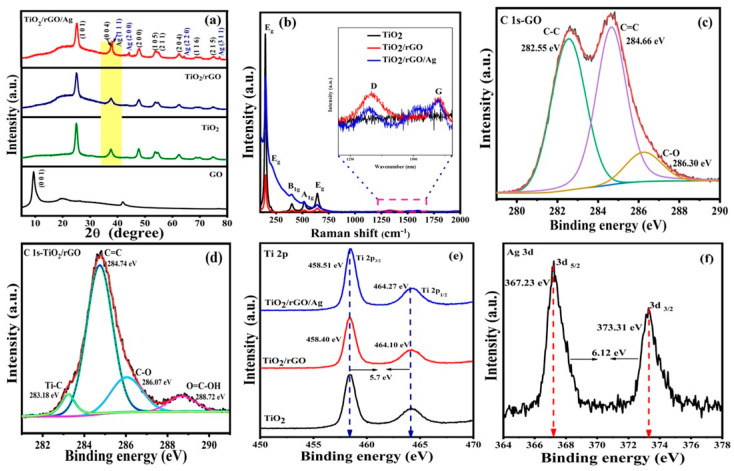
(**a**) XRD patterns, (**b**) Raman spectra of GO, mesosphere TiO_2_, TiO_2_/rGO and TiO_2_/rGO/Ag. XPS core level spectra of the (**c**) C1s spectra of GO; (**d**) C 1s spectra of TiO_2_/rGO; (**e**) Ti 2p spectra of mesosphere TiO_2_, TiO_2_/rGO and TiO_2_/rGO/Ag; (**f**) Ag 3d spectra of TiO_2_/rGO/Ag.

**Figure 2 nanomaterials-13-00065-f002:**
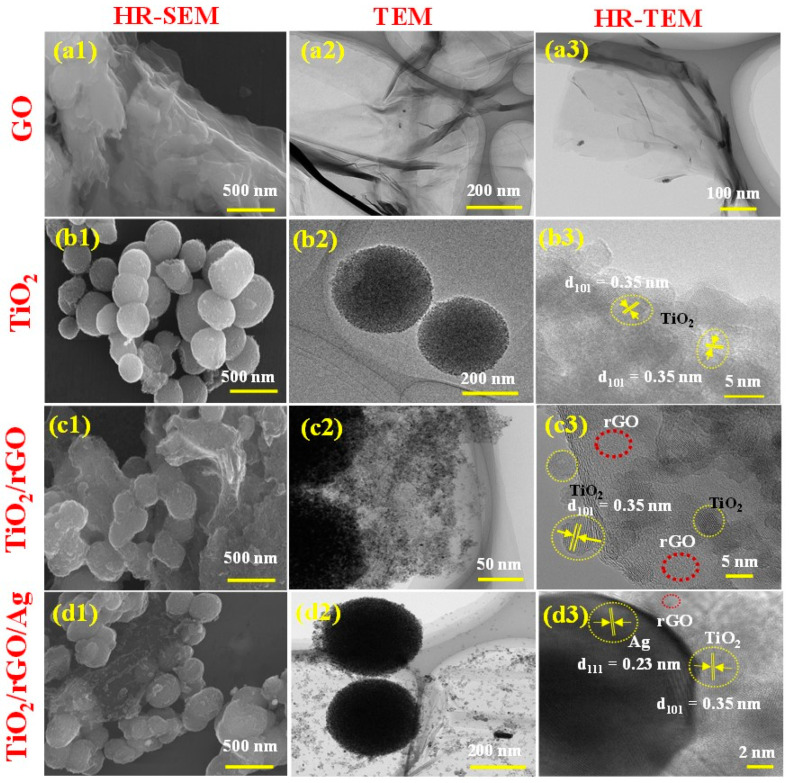
HR-SEM, TEM and HR-TEM images of prepared samples of GO (**a1**–**a3**), TiO_2_ (**b1**–**b3**), TiO_2_/rGO (**c1**–**c3**) and TiO_2_/rGO/Ag (**d1**–**d3**).

**Figure 3 nanomaterials-13-00065-f003:**
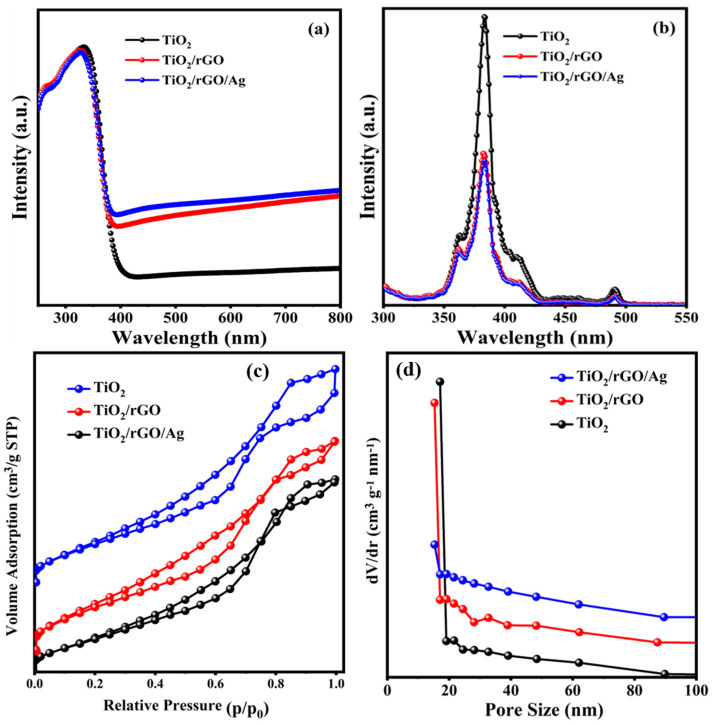
(**a**) UV-DRS absorption spectra, (**b**) PL spectra of mesosphere TiO_2_, TiO_2_/rGO and TiO_2_/rGO/Ag. Nitrogen adsorption–desorption isotherms of (**c**) surface area and corresponding (**d**) pore size distribution of prepared samples TiO_2_, TiO_2_/rGO and TiO_2_/rGO/Ag.

**Figure 4 nanomaterials-13-00065-f004:**
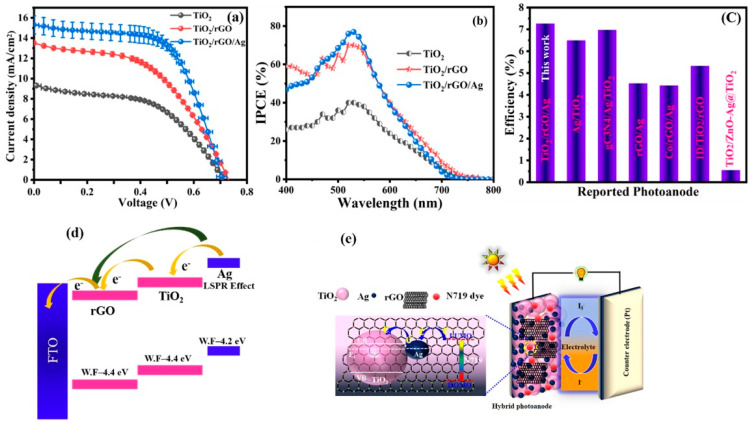
(**a**) Current density–voltage characteristics (with error bar), (**b**) IPCE spectra for different photoanode-based DSSC devices (mesosphere TiO_2_, TiO_2_/rGO and TiO_2_/rGO/Ag) measured under one sun illumination, (**c**) compared efficiency of recent reported photoanode materials with TiO_2_/rGO/Ag device performance, (**d**) schematic representation of the charge transfer process influenced by a plasmonic hybrid nanostructure (TiO_2_/rGO/Ag)-based DSSC device and (**e**) DSSC device consists of hybrid photoanode and energy level diagram.

**Figure 5 nanomaterials-13-00065-f005:**
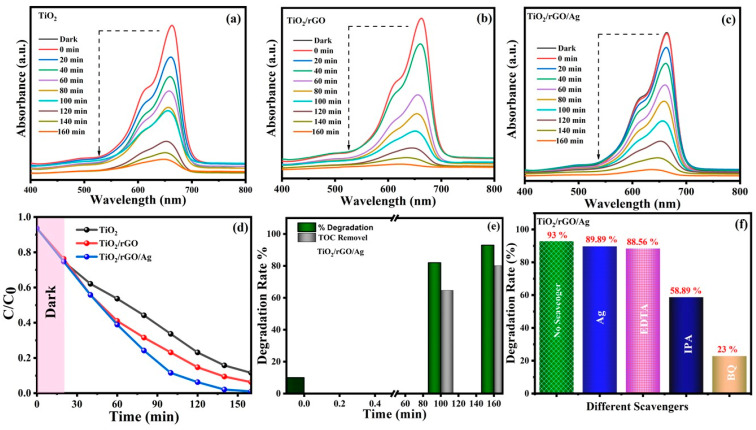
(**a**–**c**) UV absorption spectra of MB degradation with different interval time under natural sunlight, (**d**) plots of C/C_0_ as a function of time (min) towards the photo degradation of MB, (**e**) TOC analysis was observed for TiO_2_/rGO/Ag at different intervals and the (**f**) photocatalytic degradation of MB in the presence of different scavengers of the TiO_2_/rGO/Ag sample under natural sunlight.

**Figure 6 nanomaterials-13-00065-f006:**
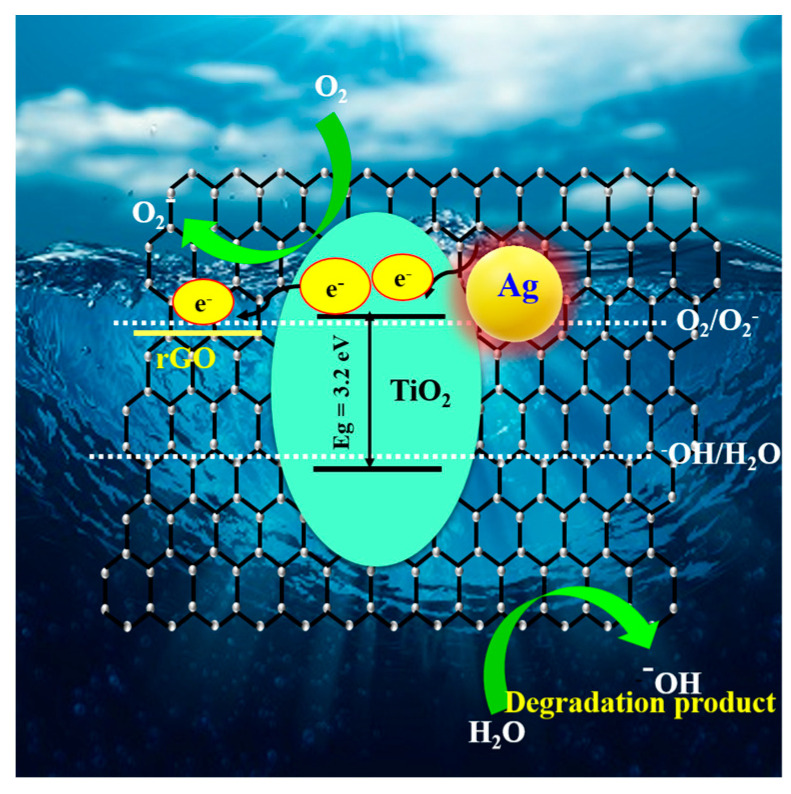
Schematic illustration of the possible mechanism of rGO, Ag NPs roles with mesosphere TiO_2_ in MB photocatalytic degradation under natural sunlight irradiation.

**Table 1 nanomaterials-13-00065-t001:** Comparison of the photovoltaic properties and surface area of the DSSC device based on the mesosphere TiO_2_, TiO_2_/rGO and TiO_2_/rGO/Ag photoanodes measured under AM 1.5 G one sun illumination.

Photoanode	Surface Area (m^2^/g)	JSC (mA/cm^2^)	VOC (V)	FF (%)	η ( %)
TiO2	234.16	9.50	0.70	51.0	4.10
TiO2/rGO	281.31	13.80	0.74	55.0	5.00
TiO2/rGO/Ag	297.71	16.05	0.74	62.5	7.27

## Data Availability

The data presented in this study are available on request from the corresponding author.

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
