# Peer review of "Plasmon Effect of Ag Nanoparticles on TiO2/rGO Nanostructures for Enhanced Energy Harvesting and Environmental Remediation"

_nanomaterials, 2022, doi:10.3390/nano13010065_

Round 1

Reviewer 1 Report

The manuscript investigates TiO2/rGO/Ag hybrid nanostructures employed as photoanodes for DSSC and catalysts for photodegradation application, resulting in an efficiency of over 7% which is 1.7 times higher than the TiO2 based DSSC.  For photocatalytic degradation of pollutants, the photocatalyst demonstrates photodegradation of methylene blue dye molecules by 93% and mineralization of total organic compounds by 80%.

The manuscript is well-organized with good-quality figures and logical scientific discussions. I believe that the manuscript will be well-appreciated by the science community. Thus, I recommend the manuscript to be accepted and published in Nanomaterials after minor text editing of the abstract.

Author Response

Respected Editor and Reviewers

We would like to thank the Editor and Reviewers for the valuable comments and suggestions to improve the quality of the manuscript. We have addressed all the comments and revised the manuscript.

Reviewer 1:

Comment 1: The manuscript is well-organized with good-quality figures and logical scientific discussions. I believe that the manuscript will be well-appreciated by the science community. Thus, I recommend the manuscript to be accepted and published in Nanomaterials after minor text editing of the abstract.

Response:

As per the reviewer suggestion, we have modified the abstract section and included in the revised manuscript as follows. (Page no: 1 line no: 16-25) (Page no: 2 line no: 26-33)

In the present report, Ag nanoparticles infused with mesosphere TiO2/reduced graphene oxide (rGO) nanosheet (TiO2/rGO/Ag) have been successfully fabricated using a series of solution-based route and in situ growth method. The prepared hybrid nanostructure is utilized for photovoltaic cells and the photocatalytic degradation of pollutants. The photovoltaic characteristics of a dye-sensitized solar cell (DSSC) device based on a photoanode with the fabricated plasmonic hybrid nanostructure (TiO2/rGO/Ag) achieved a highest short-circuit current density (JSC) of 16.05 mA/cm2, an open circuit voltage (VOC) of 0.74 V and a fill factor (FF) of 62.5%. The fabricated plasmonic DSSC device exhibited a maximum power conversion efficiency (PCE) of 7.27%, which is almost 1.7 times higher than the TiO2 based DSSC (4.10%). For the photocatalytic degradation of pollutants, the prepared TiO2/rGO/Ag photocatalyst exhibited superior photodegradation of methylene blue (MB) dye molecules at around 93% and the mineralization of total organic compounds (TOC) by 80% in aqueous solution after 160 min under continuous irradiation with natural sunlight. Moreover, the enhanced performance of the DSSC device and the MB dye degradation exhibited by the hybrid nanostructures are more associated with their high surface area and the fast transfer ability of photon-induced charge carriers with extended recombination lifetimes. Therefore, the proposed plasmonic hybrid nanostructure system is considered to be a further development for photovoltaics and environmental remediation applications.

Reviewer 2 Report

Dear authors, you have presented a very interesing hybrid nanomaterial based on TiO2 and graphene oxide. It was tested for photovoltaic cell and photocatalytic degradation under natural solar ligh.

First of all you should named all the acronims where they appears first in the text: DSSC, LSRP, rGO, CNTs...

My main concern about this paper is the part of photocatalytic degradation of methylene blue. The photocatalytic degradation was carried out under natural solar light, this is quiet good and very innovative but the problem is that you didn't measure the light irradiance and then the results are not comparable nor reproducible. The procedure isn't also well discribed, did you put your reactor perpendicular to solar light? sun is moving during the reaction, how are you sure all the experiments were carried out under similar irradiance? you should use an actinometer or calibrated solar cell.

  I mean, carring out experiments under natural solar light is really nice, but they need to be supported with experiments under controlled conditions, for instance using a solar simulator. I think this is compulsory to support the conclusions. As well as, how many repetions were carried out? at least three for every material is needed to support conclusions.

Author Response

Respected Editor and Reviewers

We would like to thank the Editor and Reviewers for the valuable comments and suggestions to improve the quality of the manuscript. We have addressed all the comments and revised the manuscript.

Reviewer 2:

Dear authors, you have presented a very interesting hybrid nanomaterial based on TiO2 and graphene oxide. It was tested for photovoltaic cell and photocatalytic degradation under natural solar light.

Comment 1: First of all you should name all the acronyms where they appears first in the text: DSSC, LSRP, rGO, CNTs...

Response:

As per the reviewer suggestion, we have mentioned the name of all the acronyms and included in the revised manuscript. (Page no: 1 line no: 20-21), (Page no: 3 line no: 65), (Page no: 1 line no: 16-17), and (Page no: 3 line no: 72).

Comment 2: My main concern about this paper is the part of photocatalytic degradation of methylene blue. The photocatalytic degradation was carried out under natural solar light, this is good and very innovative but the problem is that you didn't measure the light irradiance and then the results are not comparable nor reproducible. The procedure isn't also well described, did you put your reactor perpendicular to solar light? sun is moving during the reaction, how are you sure all the experiments were carried out under similar irradiance? you should use an actinometer or calibrated solar cell.  I mean, carring out experiments under natural solar light is really nice, but they need to be supported with experiments under controlled conditions, for instance using a solar simulator. I think this is compulsory to support the conclusions. As well as, how many repetions were carried out? at least three for every material is needed to support conclusions.

Response:

As per the reviewer suggestion, we have provided detailed experimental procedure with light intensity, latitude, and longitude of sunlight. We have performed three different trials of methylene blue dye degradation of photocatalysis and included in the revised manuscript. (Page no: 7 line no: 171-174) (Page no: 8 line no: 175-180) (Page no: 20 line no: 418-432) (Page no: 20 line no: 433-434)

Photocatalytic experiments

In all our photocatalysts measurement were carried out in our laboratory, SRM Institute of Science and Technology, Chennai (N 28o 4’N; 82o 25’E) in April 2020 (April8 to May 10). Day light from 9 am to 12 pm were utilized to perform the photocatalytic experiment. The light intensity could reach 68.2~89.4 mW/cm2.  The photocatalytic properties of the as-synthesized samples were performed using MB dye as a model pollutant under natural sunlight irradiation. In a typical photocatalytic reaction, 10 ppm of MB dye was added to 50 mL of DI water and stirred for 5 min. To achieve an adsorption-desorption equilibrium, the solution was maintained in the dark for 20 min under stirring. At regular time intervals of the photocatalytic dye degradation reaction solution (20 min), 3 mL aliquots of the solution were sampled and UV-Vis spectra were measured.

To demonstrate the photocatalytic MB dye degradation process, TiO2, TiO2/rGO, and TiO2/rGO/Ag based photocatalysts were employed under natural sunlight irradiation. Figure 1(a-c) shows three different trials of dye degradation profiles with MB as the pollutant with different irradiation times under natural sunlight for all the prepared photocatalysts. The maximum absorption peak for the photocatalytic dye degradation process in the presence of a photocatalyst is displayed at around 664 nm due to the absorbance characteristics of MB dye. Under dark conditions, there were no significant changes in the UV-vis absorption spectra of all the prepared photocatalysts. UV-vis absorption spectra of all the prepared photocatalysts were successfully recorded with a time interval of 20 min with continuous natural sun irradiation. The hybrid nanostructures of the TiO2/rGO/Ag photocatalyst of all the three exhibited a higher dye degradation efficiency of 93% under continuous natural sunlight irradiation for 160 min. Compared with pristine TiO2, the enhanced dye degradation efficiency of the plasmonic hybrid photocatalyst was almost 1.3 times higher due to the LSPR properties of the Ag NPs. On the other hand, the presence of rGO/Ag on the TiO2 surface creates a more favourable environment for fast photon induced charge separation and transfer with an extended lifetime of charge carriers, whereas Ag NPs act as electron sinks for improved photocatalytic dye degradation [1]. Figure 2 shows the C/C0 graph with respect to time which shows the plasmonic enriched Ag NPs exhibiting excellent photodegradation at each interval time, which comparably higher than TiO2 and TiO2/rGO sample.

Reference:

[1]. Y. He, P. Basnet, S.E.H. Murph, Y. Zhao, Ag nanoparticle embedded TiO2 composite nanorod arrays fabricated by oblique angle deposition: Toward plasmonic photocatalysis, ACS Appl. Mater. Interfaces. 5 (2013) 11818–11827. https://doi.org/10.1021/am4035015.

Fig. S8: (a-c) Three different trials of UV absorption spectra of MB degradation with different interval time under natural sunlight.

Reviewer 3 Report

Review Comments: The authors of the manuscript fabricated hybrid nanostructures of Ag nanoparticles infused with mesosphere TiO2/rGO nanosheets following a series of solution processes synthesis routes and in-situ growth for photovoltaic cell and photocatalytic degradation. They applied the plasmonic nanostructure in the photoanode of dye-sensitized solar cells (DSSCs) and photocatalytic degradation of methylene blue as a photocatalyst. The work is scientifically important and within the scope of the journal; however, the manuscript cannot be accepted in its current form. The authors may be advised to improve their manuscript according to the following comments.

1.      Line # 34-35, the authors mentioned “The remarkable increase in energy need and depletion of fossil fuels urge researcher to find out energy harvesting from non-renewable energy resources”. Did the authors mean to say that fossil fuels are renewable energy resources?

2.      Line # 68, Where is Eid et al.? There is no such author in the cited articles.

3.      References 22 and 24 are not appropriate.

4.      Line # 96, the authors mentioned LSPR without definition.

5.      Line # 101-103 & 132, the authors used abbreviation and formula randomly. If they used “HCL” as the chemical formula of hydrochloric acid, then they should use “HCl” instead of “HCL” that is “Cl” for chlorine.

6.      Thickness of the photoactive layer is an important factor in determining the performance of DSSCs. The authors should show thickness profiles of TiO2, TiO2/rGO, and TiO2/rGO/Ag.

7.      In Figure 5, the authors did not show the corresponding equivalent circuit. Without defining appropriate equivalent circuit, the readers may find it hard to understand the meaning of Rs and Rct.

8.      The authors measured the impedance spectra from 10 mHz to 1000 kHz at -0.8 V. From the statement it is not clear what the amplitude of the AC perturbation was. Also, according to the current-voltage curves (Figure 4a), it is evident that the DSSCs at -0.8 V (other than at VOC) must showed different current. That means comparing such impedance data does not make any sense.   

9.       The authors did not use term “Figure” or “Fig.” consistently throughout the manuscript.

10.   Line # 478, the “x” is subscripted.

11.  The authors mentioned “synergistic effect of Ag nanoparticles” in the title but they did not verify that or even mention the term “synergistic” in their manuscript.

Author Response

Respected Editor and Reviewers

We would like to thank the Editor and Reviewers for the valuable comments and suggestions to improve the quality of the manuscript. We have addressed all the comments and revised the manuscript.

Reviewer 3:

Comment 1: Line # 34-35, the authors mentioned “The remarkable increase in energy need and depletion of fossil fuels urge researcher to find out energy harvesting from non-renewable energy resources”. Did the authors mean to say that fossil fuels are renewable energy resources?

Response:

As per the reviewer suggestion, we have modified the sentence and included in the manuscript as follows. (Page no:2 line:37-38)

            The significant increase in energy requirements and the depletion of fossil fuels have urged researchers to develop energy harvesting from renewable energy resources [1,2].

Reference:

[1]. F. Jamil, H.M. Ali, M.M. Janjua, MXene based advanced materials for thermal energy storage: A recent review, J. Energy Storage. 35 (2021) 102322. https://doi.org/10.1016/j.est.2021.102322.

[2]      A. Sözen, Ç. Filiz, İ. Aytaç, K. Martin, H.M. Ali, K. Boran, Y. YetiÅŸken, Upgrading of the Performance of an Air-to-Air Heat Exchanger Using Graphene/Water Nanofluid, Int. J. Thermophys. 42 (2021). https://doi.org/10.1007/s10765-020-02790-w.

Comment 2: Line # 68, Where is Eid et al.? There is no such author in the cited articles.

Response:

We apologise for the typo. As per the reviewer suggestion, we have modified the appropriate reference and included in the revised manuscript. (Page no: 3 line: 71-74)

Reference:

[21]    D. Zhao, X. Yang, C. Chen, X. Wang, Journal of Colloid and Interface Science Enhanced photocatalytic degradation of methylene blue on multiwalled carbon, 398 (2013) 234–239. https://doi.org/10.1016/j.jcis.2013.02.017.

Comment 3: References 23 and 25 are not appropriate.

Response:

As per the reviewer suggestion, we have modified the appropriate references and included in the revised manuscript. (Page no: 4 line: 77-81)

Reference:

[23]    S.M. Parsa, A. Yazdani, H. Dhahad, W.H. Alawee, S. Hesabi, F. Norozpour, D. Javadi Y, H.M. Ali, M. Afrand, Effect of Ag, Au, TiO2 metallic/metal oxide nanoparticles in double-slope solar stills via thermodynamic and environmental analysis, J. Clean. Prod. 311 (2021) 127689. https://doi.org/10.1016/j.jclepro.2021.127689.

[25]    P. Wang, J. Wang, X. Wang, H. Yu, J. Yu, M. Lei, Y. Wang, Applied Catalysis B : Environmental One-step synthesis of easy-recycling TiO 2 -rGO nanocomposite photocatalysts with enhanced photocatalytic activity, "Applied Catal. B, Environ. 132–133 (2013) 452–459. https://doi.org/10.1016/j.apcatb.2012.12.009.

Comment 4: Line # 96, the authors mentioned LSPR without definition.

Response:

As per the reviewer suggestion, we have mentioned the definition of LSPR and included in the revised manuscript. (Page no: 3 line no: 65-67).

Comment 5:  Line # 101-103 & 132, the authors used abbreviation and formula randomly. If they used “HCL” as the chemical formula of hydrochloric acid, then they should use “HCl” instead of “HCL” that is “Cl” for chlorine.

Response:

As per the reviewer suggestion, we have modified “HCL” to “HCl” and included in the revised manuscript. (Page no: 5 line no: 104)

Comment 6:   Thickness of the photoactive layer is an important factor in determining the performance of DSSCs. The authors should show thickness profiles of TiO2, TiO2/rGO, and TiO2/rGO/Ag.

Response:

As per the reviewer suggestion, we have performed the thickness profiles of TiO2 and TiO2/rGO/Ag and included the discussion in the revised manuscript. (Page no: 13 line no: 293-294)

The fabricated TiO2 and TiO2/rGO/Ag photoanode were used to calibrate the thickness using HR-SEM analysis. The thickness of the fabricated photoanode was estimated to be 14.2 µm (TiO2) and 13.8 µm (TiO2/rGO/Ag) and presented Fig. S5(a-b).

Fig S5: HR-SEM cross section image of TiO2 and TiO2/rGO/Ag

Comment 7: In Figure 5, the authors did not show the corresponding equivalent circuit. Without defining appropriate equivalent circuit, the readers may find it hard to understand the meaning of Rs and Rct.

Response:

As per the reviewer suggestion, we have included the equivalent circuit as an inset image in Fig. 5(a) and included the discussion in the revised manuscript. (Page no: 18 line no:383-391) (Page no: 19 line no:392-402)

Fig 5: (a) Nyquist plot (b) Extended Nyquist plot plots obtained for fabricated device of mesosphere TiO2, TiO2/rGO, TiO2/rGO/Ag and the equivalent circuit of the device.

Nyquist plots for all fabricated DSSC devices measured in the frequency range of 10 mHz to 1000 kHz at an applied bias of -0.8 V under dark conditions are shown in Fig. 5(a) and fitted value presented in Table. S1. The observed experimental Nyquist plots were fitted with equivalent circuits; scattered points and lines represent the experimental and fitted data, respectively (Fig. 5(b)). The Nyquist plot of the TiO2/rGO/Ag photoanode based DSSC device has a small semicircle arc, which indicates low charge transfer resistance (Rct). However, the higher and lower frequency response regions in the observed semicircle represent the series resistance (Rs) and charge transfer resistance (Rct) at the photoanode/dye/electrolyte interface. The lower series resistance of the device (Rs: 21.94 Ω) in the case of the TiO2/rGO/Ag based photoanode indicates reduced ohmic resistance in series with the photoanode and an external device due to the superior electrical conductivity of the plasmonic DSSC device compared to other devices. Extended views of the Nyquist plots at higher frequency for all the fabricated DSSC devices are given in Fig. 5b. The lower charge transfer resistance (Rct: 26.56 Ω) after introduction of the rGO/Ag layer on the TiO2 surface reveals the charge transfer characteristics at the photoanode/dye/electrolyte interface, where the rGO/Ag layer acts as the electrocatalytic layer [62]. The overall EIS measurements of fabricated devices are well correlated with the enhanced photovoltaic characteristics, as shown in Fig. 4. Therefore, the plasmonic hybrid photoanode enables fast charge transfer with a prolonged electron lifetime to the FTO substrate in a DSSC device for enhanced photovoltaic performance.

Comment 8: The authors measured the impedance spectra from 10 mHz to 1000 kHz at -0.8 V. From the statement it is not clear what the amplitude of the AC perturbation was. Also, according to the current-voltage curves (Figure 4a), it is evident that the DSSCs at -0.8 V (other than at VOC) must showed different current. That means comparing such impedance data does not make any sense.

Response:

We agree with the reviewers, but the typical Electrochemical Impedance Spectroscopy (EIS) technique is often identified as an AC (Alternating Current) technique whereas, the sinusoidal stimulation waveform has only used as an input single. As compared with EIS, the other techniques are identified as DC (Direct Current) techniques such as LSV, CV and so on. Moreover, With AC techniques, such as Electrochemical Impedance Spectroscopy (EIS), the response of the system to a potential or current sinusoidal perturbation is studied as a function of the frequency, which is swept over a few decades as higher to low frequencies with help of small AC sinusoidal perturbation potential 5 mV. All the points and validations are common for EIS analysis according to Bio logic standard electrochemical workstation.

Yes, EIS technique is a more powerful technique for analyse the electrode/electrolyte electronic interface properties such as charge transfer resistance, sheet resistance, lifetime of charge carriers at electrode/electrolyte interface. In our present work, we have carried out the Nyquist plot of fabricated DSSC system using open circuit potential used as applied external varying with respect to applied bias which depends upon our device. Moreover, our ultimate aim has to analysis impedance at interface under unbiased conditions nearly open circuit potential as in built photovoltage. As shown J-V profile Fig. 4(a), different photocurrent density offered with respect to potential window of fabricated DSSC devices owing to possesses charge transfer resistance. For our continence and comparison, same Voc value has used to all other DSSC devices as an applied external bias which has clearly explored the impedance characterization with some operating conditions.   

Fig 5: (a) Nyquist plot (b) Extended Nyquist plot plots obtained for fabricated device of mesosphere TiO2, TiO2/rGO, TiO2/rGO/Ag and the equivalent circuit of the device.

Comment 9: The authors did not use term “Figure” or “Fig.” consistently throughout the manuscript.

Response:

As per the reviewer suggestion, we have used term “Figure” to “Fig” and included in the revised manuscript.

Comment 10: Line # 478, the “x” is subscripted.

Response:

As per the reviewer suggestion, we have modified and included in the revised manuscript. (Page no:478 line no:20)

Comment 11: The authors mentioned “synergistic effect of Ag nanoparticles” in the title but they did not verify that or even mention the term “synergistic” in their manuscript.

Response:

As per the reviewer suggestion, we have modified the title as “Plasmonic effect of Ag nanoparticles on TiO2/reduced graphene oxide nanostructures for enhanced charge transfer process in energy harvesting and environmental remediation” and included in the revised manuscript.

Reviewer 4 Report

The work is well and thoroughly described and will be of interest to the broad readership of Nanomaterials. When reading, the work is tiring with a large number of abbreviations. Apparently, this is why the authors did not decipher several of them, i.e. DSSC, rGO, LSPR. Also, in some places, there are no gaps between sentences in the text.

Author Response

Respected Editor and Reviewers

We would like to thank the Editor and Reviewers for the valuable comments and suggestions to improve the quality of the manuscript. We have addressed all the comments and revised the manuscript.

Reviewer 2:

Comment 1: The work is well and thoroughly described and will be of interest to the broad readership of Nanomaterials. When reading, the work is tiring with a large number of abbreviations. Apparently, this is why the authors did not decipher several of them, i.e. DSSC, rGO, LSPR. Also, in some places, there are no gaps between sentences in the text.

Response:

As per the reviewer suggestion, we have modified the decipher term in the revised manuscript. (Page no: 1 line no: 20-21), (Page no: 1 line no: 16-17) and (Page no: 3 line no: 65)

Round 2

Reviewer 2 Report

Thank you for taking into account all my recomendations

Author Response

Respected Editor and Reviewer

We would like to thank the Editor and Reviewer for the valuable comments and suggestions to improve the quality of the manuscript. We have addressed all the comments and revised the manuscript.

Reviewer 2:

Comment 1: Thank you for taking into account all my recommendations

Response:

We thank the reviewer for recommending our manuscript.

.

Reviewer 3 Report

The authors have updated their manuscript according to my comments; however, the impedance analysis part (Section 3.5 & Figure 5) is inappropriate and contains serious flaws. There are lots of articles on the equivalent circuits and impedance analysis of DSSCs. The authors should consult those articles and update Section 3.5 accordingly. One such article can be found here https://www.hindawi.com/journals/ijp/2014/851705/

Author Response

Respected Editor and Reviewer

We would like to thank the Editor and Reviewer for the valuable comments and suggestions to improve the quality of the manuscript. We have addressed all the comments and revised the manuscript.

Reviewer 3:

Comment 1: The authors have updated their manuscript according to my comments; however, the impedance analysis part (Section 3.5 & Figure 5) is inappropriate and contains serious flaws. There are lots of articles on the equivalent circuits and impedance analysis of DSSCs. The authors should consult those articles and update Section 3.5 accordingly.

Response:

As per the reviewer suggestion, we have modified the impedance analysis part (Section 3.5 & Figure 5) and included in the revised manuscript. (Page no: 18 line no: 380-391) (Page no: 19 line no: 392-410)

Electrochemical impedance spectroscopy (EIS) analyses were performed to determine the interface charge transfer dynamics and recombination lifetime of charge carrier related characteristics of the fabricated DSSC device. Fig. 5(a) and 5 (b) shows the Nyquist plot of all fabricated DSSC devices. The EIS analysis was measured in the frequency range of 10 mHz to 1000 kHz at an applied bias of VOC under dark conditions. The obtained Nyquist plots were fitted with an equivalent circuit and experimental and fitted data were represented using scattered point and solid lines respectively (as shown in Fig. 5(b)). Typically, there are two semicircles were observed in the Nyquist Plot from low frequency to high frequency region related to the faraday resistance of redox reaction and I-3/I- at the counter electrode (Pt)/electrolyte interface and charge transfer resistance at TiO2/electrolyte/dye interface (Rct) respectively [1]. Further, the intercept of the high frequency region on x-axis denoted as series resistance (Rs) of the fabricated DSSC system. Based on the obtained Rct result, charge transfer resistance of TiO2 is decreased from 25.67 Ω to 21.94 Ω with the addition of rGO in TiO2 network. This result reveals that the rGO induced the charge transfer among the interfaces and also increase the electron lifetime that hinder recombination of photoexited charge carrier and accelerates the charge transportation is benefiting to DSSC performance. Also, TiO2-rGO exhibited the small Rs value infers that improved electrical conductivity and relatively easy redox reaction [2]. In the case of TiO2/rGO/Ag, at the low frequency region the radius of the semicircle arc further decreased due to the enhancement of electrolyte diffusion. This may be attributed to the fact that the Ag may also act as scattering points in the photoanode system (Plasmonic hybrid structure). This results further suggested that while introducing Ag the charge transfer resistance has decreased (21.94 Ω) as compared to TiO2 (25.67 Ω) and TiO2/rGO (22.37 Ω) hence, Ag can also accelerate electron transportation [3-5]. Moreover, the presence of rGO/Ag as a modified layer is critical factor to reduce the charge transfer resistance with extending lifetime of photon induced charge carriers via forming an effective charge transfer channel. It has been extremely beneficial to improve their photogenerated charge carrier collection rate [6]. This is more convincing with the improved Jsc factor of their DSSC device as well as observed boosted photovoltaic characteristics performance as shown in Fig.4. Thus, Plasmonic hybrid photoanode demonstrates that it facilitates the fast charge transfer of photoexcited charge carrier and accelerates electron transport in DSSC device for enriched photovoltaic performance.     

Fig 5: (a) Nyquist plot (b) Extended Nyquist plot plots obtained for fabricated device of mesosphere TiO2, TiO2/rGO, TiO2/rGO/Ag and the equivalent circuit of the device.

Reference:.

   4         M. Kandasamy, M. Selvaraj, C. Kumarappan, S. Murugesan, Plasmonic Ag       nanoparticles anchored ethylenediamine modified TiO2 nanowires@graphene oxide composites for dye-sensitized solar cell, J. Alloys Compd. 902 (2022) 163743. https://doi.org/10.1016/j.jallcom.2022.163743.

 5        I. Ahmad, R. Jafer, S.M. Abbas, N. Ahmad, Ata-ur-Rehman, J. Iqbal, S. Bashir, A.A. Melaibari, M.H. Khan, Improving energy harvesting efficiency of dye sensitized solar cell by using cobalt-rGO co-doped TiO2 photoanode, J. Alloys Compd. 891 (2022) 162040. https://doi.org/10.1016/j.jallcom.2021.162040.

Round 3

Reviewer 3 Report

The manuscript is still not publishable in its current form. The authors must consider the following comments to update their manuscript.

1. The title of the manuscript and the supporting information are not  the same.

2. The impedance analysis part (Section 3.5 & Figure 5) is l inappropriate and contains serious flaws.  Typically DSSCs show three semi-circles not two. The equivalent circuit shown in the inset of the impedance spectra can fit only one semi-circle. That means, Rct values obtained by fitting the impedance spectra to the equivalent circuit  were not correctly evaluated, Therefore, the conclusion of the analysis is not acceptable. For a better understanding of the impedance analysis of DSSCs, the authors may consult this article: https://www.hindawi.com/journals/ijp/2014/851705/

Author Response

Respected Editor and Reviewer

We would like to thank the Editor and Reviewer for the valuable comments and suggestions to improve the quality of the manuscript. We have addressed all the comments in the revised manuscript.

Reviewer 3:

Comment 1:  The title of the manuscript and the supporting information are not the same.

Response

As per the reviewer suggestion, we have modified the title in the Supporting information and included in the revised manuscript.

Comment 2:  The impedance analysis part (Section 3.5 & Figure 5) is l inappropriate and contains serious flaws.  Typically, DSSCs show three semi-circles not two. The equivalent circuit shown in the inset of the impedance spectra can fit only one semi-circle. That means, Rct values obtained by fitting the impedance spectra to the equivalent circuit were not correctly evaluated, Therefore, the conclusion of the analysis is not acceptable. For a better understanding of the impedance analysis of DSSCs, the authors may consult this article: https://www.hindawi.com/journals/ijp/2014/851705/

Response

As per the reviewer suggestion, we have tried our best to record the impedance spectra, however due to unavailability of instrument we could not able to perform the analysis in the given deadline. Hence, we have decided to remove the EIS part in the revised version. Thanks for the comments to improve our manuscript.
